# Tokens that Know Where: Self-improving 2D Spatial Vocabulary for Multi-modal Understanding

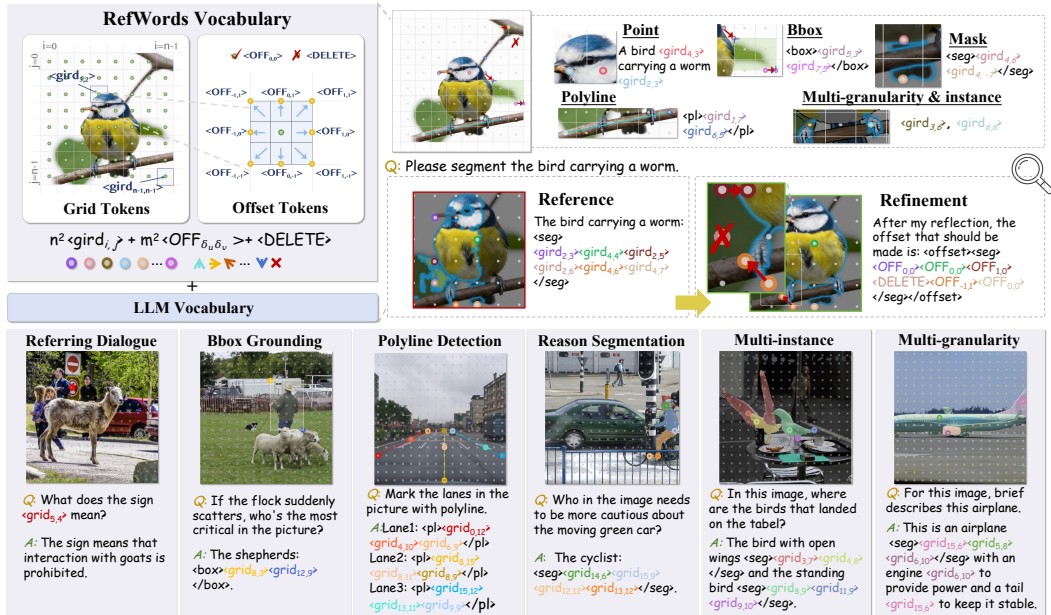

Figure 1: **Top:** Overview of *RefWords*, a vocabulary of pre-defined, learnable discrete tokens for 2D spatial representation. It comprises *grid tokens* anchored to the image plane and *offset tokens* for iterative refinement. **Bottom:** *RefWords* support both referring inputs and outputs; multiple format conversion such as boxes, polylines, and masks; and seamless compatibility with multi-instance and multi-granularity. No additional components need to be trained. Best viewed in color.

## Abstract

Due to the inherent loss of spatial information caused by token serialization in autoregressive frameworks, modern multimodal large language models (MLLMs) continue to encounter significant challenges in understanding and accurately referencing 2D spatial locations. In this work, we address a critical question: *How can sequential tokens create a learnable and robust mapping to continuous 2D spatial positions?* We introduce RefWords, a spatial representation that integrates a dedicated vocabulary of learnable tokens into MLLMs. RefWords is featured by two key components: (1) Grid Tokens, which divide the image plane into structured spatial anchors, and (2) Offset Tokens, which allow for detailed, iterative refinement of localization predictions. By embedding spatial relationships directly into the token representation space, RefWords enables MLLMs to perform native 2D reasoning without altering the autoregressive architecture. Extensive experiments demonstrate that RefWords achieves superior performance across various referring tasks in both supervised and reinforcement learning settings. This shows that sequential tokens can effectively represent 2D space when provided with structured representations. This work presents a new paradigm for spatial reasoning in multi-modal systems.

# 1 INTRODUCTION

The impressive success of autoregressive Transformers in language modeling has established them as the core architecture for multimodal large language models (MLLMs). This has significantly advanced applications such as multimodal dialogue systems and web agents. However, their deployment in real-world scenarios, such as autonomous driving and embodied control, is still limited due to challenges with accurate spatial understanding and reliable location referencing. We recognize that a key limitation affecting the spatial understanding of MLLMs is the gap between textual tokens and continuous visual regions. This fundamental misalignment arises from the conflict between sequential processing of tokens and the modeling of 2D spatial information, resulting in consistent failures during precise localization tasks.

Existing approaches address the representational gap through compromise. Coordinate-based methods represent locations as text strings; for instance, a bounding box might be represented as $(x_1, y_1, x_2, y_2)$. However, this method does not preserve spatial topology. For example, coordinates that are spatially close, like `"199"` and `"200"`, may be distant in the textual token space, differing by three characters. Moreover, these representations are inefficient due to syntactic overhead, which includes commas and brackets, and they show fragility to minor token errors. Alternative methods use specialized components, such as regional encoders or segmentation heads, to address these issues. However, these approaches abandon the unified sequence-to-sequence paradigm, which introduces architectural complexity and limits generalization across various referring tasks.

To improve the ability of MLLMs to understand and refer to 2D spatial information, this work tackles a fundamental issue: how can sequential tokens be designed to inherently represent and reason about 2D space within an unchanged autoregressive architecture? The primary challenge is to create a reliable mapping between discrete tokens and continuous 2D space—one that maintains spatial topology while remaining compatible with autoregressive generation. As such, we propose *RefWords*, a lexicalized 2D spatial representation that addresses this challenge through a set of learnable *spatial vocabulary*. RefWords is featured by two core components:

i) **Grid Tokens** establish a structured spatial topology by discretizing the image plane into an $n \times n$ uniform grid. Each grid cell is associated with a learnable token added to the model's vocabulary, creating a set of *spatial anchors* where each token is responsible for referring to objects within its corresponding local region.

ii) **Offset Tokens** enable precise spatial refinement by introducing an $m \times m$ set of discrete displacement vectors (`<OFF`$_{-1,1}$`>`,...,`<OFF`$_{0,0}$`>`,..., `<OFF`$_{1,-1}$`>`) together with a `<DELETE>` token. Building upon the structural regularity of grid tokens, the offset mechanism enables local adjustments to initial predictions, forming an iterative refinement process that mimics human visual localization. This creates a *self-improving* localization system, where `<DELETE>` provides built-in rejection capability, establishing a cohesive feedback loop for introspective refinement and error correction.

RefWords presents three key advantages compared to existing methods: First, it replaces the unreliable practice of using text strings to describe locations with a dedicated spatial vocabulary. This approach embeds two-dimensional spatial relationships directly into the token representation space, allowing for native spatial reasoning. Second, RefWords offers a unified representation for various tasks—ranging from points to masks—entirely within a standard autoregressive framework. This eliminates the need for task-specific modules, simplifying the architecture while maintaining generalizability and precision. Third, the integrated offset mechanism enables self-correction through iterative refinement. This allows the model to critique and adjust its spatial predictions, a feature lacking in existing methods where initial errors often remain uncorrected.

The main contributions are summarized as follows:

- We propose RefWords, a lexical spatial representation that embeds a vocabulary of learnable tokens into the model, enabling native 2D spatial reasoning within unmodified autoregressive frameworks.

- We introduce an introspective refinement mechanism that creates a self-improvement loop, resolving error irreversibility in existing methods.

- We develop training methodologies for both supervised fine-tuning and reinforcement learning, including automated data transformation and specialized reward functions, demonstrating RefWords' versatility across learning paradigms.

Extensive experiments on diverse referring benchmarks show that RefWords achieves superior performance under both supervised fine-tuning and reinforcement learning. Our work demonstrates the feasibility and advantages of modeling 2D spatial information through structured token representations, paving the way for a novel approach to spatial reasoning in multimodal systems.

## 2 REFERRING REPRESENTATIONS IN VISION–LANGUAGE MODELS

Enabling multimodal large language models to achieve precise spatial understanding and referring remains challenging. Current approaches typically follow several distinct paradigms.

**Coordinate-Based Representations.** Methods like Shikra Chen et al. (2023), GPT4RoI Zhang et al. (2023) represent spatial locations as textual coordinate strings. While this approach is straightforward, it has fundamental limitations. The issue of metric-token mismatch disrupts spatial topology, while syntactic overhead reduces token efficiency, and format brittleness compromises robustness.

**Discretized Coordinate Methods.** Approaches such as Pix2Seq Chen et al. (2021), OFA Wang et al. (2022a) attempt to mitigate these issues through coordinate discretization into bin tokens. However, they still treat localization as a numerical prediction task, requiring multiple tokens per coordinate and failing to capture the semantic nature of spatial relationships. Kosmos-2 Peng et al. (2023) represents a step toward spatial grounding but remains limited to bounding box tasks without supporting the full spectrum of referring capabilities.

**Specialized Architectural Components.** Methods like LISA Lai et al. (2024) and Ferret You et al. (2023) incorporate task-specific modules such as regional encoders or segmentation heads. While effective for particular tasks, these approaches abandon the unified sequence-to-sequence paradigm, introducing architectural complexity and limiting generalization across diverse referring scenarios.

**Reinforcement Learning for Localization.** Recent work like Seg-Zero Liu et al. (2025a) explores reinforcement learning for visual grounding, often decoupling reasoning from segmentation through prompt-based mechanisms. While achieving impressive results, these methods still encode locations as *textual coordinates* rather than native spatial understanding within the language model itself.

Unlike these approaches, RefWords introduces a fundamentally different paradigm: rather than adapting spatial information to fit existing token representations or adding specialized components, we propose a native spatial vocabulary that operates within standard autoregressive frameworks. This lexicalized approach maintains architectural unity while enabling precise 2D spatial reasoning across diverse referring tasks.

## 3 REFERRING VLMS WITH REFWORDS

We introduce *RefWords*, a unified referring representation designed to endow LVLMs with the ability to interpret and generate spatial references in a native token-based manner. The core idea is to augment the model's vocabulary with a set of learnable spatial tokens—*Grid Tokens* and *Offset Tokens*—that collectively form a structured spatial lexicon. This vocabulary enables the model to express locations through a cohesive *propose-and-refine* chain, seamlessly integrated into standard autoregressive decoding.

**RefWords Vocabulary.** As illustrated in Figure 1, RefWords introduces a unified spatial vocabulary comprising two complementary token types that enable native visual referencing in LVLMs. *i) Grid Tokens* discretize the image into $n \times n$ anchors: $\mathcal{T}_{\text{grid}} = \{\texttt{<grid}_{i,j}\texttt{>} \mid i,j \in \{0,\ldots,n-1\}\}$. *ii) Offset Tokens* enable local refinement: $\mathcal{T}_{\text{offset}} = \{\texttt{<OFF}_{\delta_u,\delta_v}\texttt{>}\} \cup \{\texttt{<DELETE>}\}$, where $\delta_u, \delta_v \in \{-1, 0, 1\}$. The complete vocabulary $\mathcal{V} = \mathcal{V}_{\text{LLM}} \cup \mathcal{T}_{\text{grid}} \cup \mathcal{T}_{\text{offset}}$ facilitates spatial reasoning as precise *spatial pronouns*. Coordinate mappings are detailed in the appendix.

**Training and Inference.** We implement RefWords under two complementary training paradigms: *Supervised Fine-Tuning (SFT)* and *Reinforcement Learning (RL)*. Figure 2 illustrates the core

propose-and-refine mechanism of RefWords, where grid tokens provide initial coarse localization and offset tokens enable precise spatial adjustments through iterative refinement. For SFT, we focus on constructing training data that enables the model to properly utilize the RefWords vocabulary through automated annotation conversion and sequence simulation. For RL, we adapt the GRPO Shao et al. (2024) framework with specialized reward functions designed for referring expression segmentation, guiding the model to generate spatially accurate and well-formed references. During inference, the generated token sequences are directly converted back to spatial coordinates using mapping rules, maintaining text-based simplicity while achieving precise spatial grounding. The following two sections detail our approaches for SFT and RL training respectively.

## 4 REFWORDS-SFT: REFWORDS FOR SUPERVISED FINE-TUNING

The integration of RefWords into a SFT paradigm provides a straightforward yet effective approach to endow MLLMS with spatial reasoning capabilities. The core premise treats spatial reference generation as a standard sequence modeling task within the autoregressive framework.

Figure 2: An overview of the propose-and-refine mechanism in RefWords with a result example. Grid tokens provide coarse localization, while offset tokens enable precise adjustment.

### 4.1 GRID TOKEN DATA CURATION

In this section, we show how existing referring-related datasets can be converted into the unified grid token representation.

**Conversion from/to Point, Bounding Boxes and Polylines.** Grid tokens establish a unified representation for diverse geometric forms through straightforward computational mappings. Points are directly represented by their nearest grid token. Bounding boxes are encoded as `<box><grid_{i_1,j_1}><grid_{i_2,j_2}><\box>`, where the two tokens correspond to the top-left and bottom-right corners identified through nearest-grid assignment. Similarly, polylines—commonly used in applications such as lane detection and motion planning—are represented as ordered sequences of grid tokens approximating the curve nodes, using the format `<pl><grid_{i_1,j_1}>...<grid_{i_n,j_n}><\pl>`.

**Greedy Mask-to-Token Conversion Algorithm.** Segmentation masks are a type of referring representation that is much finer and more accurate. Unlike previous formats, its continuous nature makes it tricky to discretize using grid tokens.

For the conversion from grid tokens to masks, we can easily implement this using off-the-shelf SAM by using grid tokens as point prompts. The more challenging aspect is the reverse process: how to convert masks into grid tokens? Naive approaches such as using single points, bounding boxes, or randomly sampled points within a mask suffer from redundancy and ambiguity, particularly in the case of complex, multiply-connected mask regions. We design a greedy algorithm to facilitate the transformation from masks to grid tokens. Notably, this conversion requires no training and can be viewed as a straightforward data processing step, thereby avoiding any unnecessary complexity in our architecture.

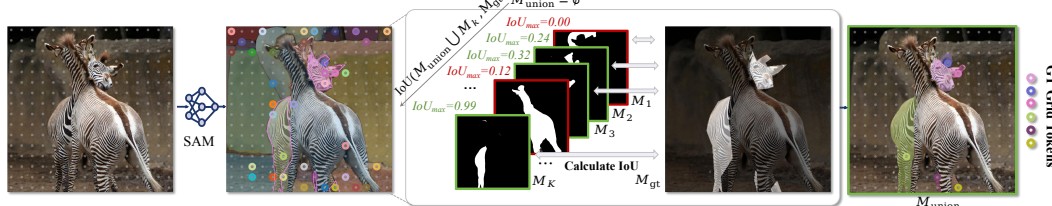

Figure 3: **The greedy algorithm for generating the ground-truth grid tokens sequence corresponding to the ground-truth mask.** This conversion automatically transforms continuous masks into discrete tokens, enabling scalable data expansion.

Specifically, we first input the image, together with the $n^2$ grid points as prompts into SAM[1], produces $K$ masks $\mathcal{M} = \{\mathbf{M}_1, \ldots, \mathbf{M}_K\}$. Each mask uniquely corresponds to an input grid, denoting by a mapping $\theta : \{i\}_{i=1}^{n^2} \to \{k\}_{k=1}^{K}$. Typically, $K < n^2$ due to the mask deduplication that occurs in post-processing. Given a ground-truth mask $\mathbf{M}_{\text{gt}}$, we aim to find *a minimal set of* grid points so that the union of their corresponding masks resembles $\mathbf{M}_{\text{gt}}$. The objective can be written as:

$$\boldsymbol{\pi}^* = \arg\min_{\boldsymbol{\pi}} \|\boldsymbol{\pi}\|_1$$
$$\text{s.t.} \quad \text{IoU}(\mathbf{M}_{\text{gt}}, \bigcup_{k:\pi_k=1} \mathbf{M}_{\theta(k)}) \geq \tau, \tag{1}$$

where $\boldsymbol{\pi} \in \{0,1\}^{n^2}$ is a binary selection vector over grid tokens, and $\tau$ is a quality threshold ensuring a minimum Intersection-over-Union (IoU). Equation 1 is a constrained multi-object optimization problem. For efficiency, we design a simple greedy algorithm and find it suffices for a decent solution. The algorithm begins with $\boldsymbol{\pi} = \mathbf{0}$, $\mathbf{M}_{\text{union}} = \mathbf{0}$, and $\text{IoU}_{\text{max}} = 0$. We first compute IoUs between $\mathbf{M}_{\text{gt}}$ and all $K$ mask proposals, and sort them in descending order. We then iterates through all masks, for the $k$-th iteration, we calculate $\text{IoU}^* = \text{IoU}(\mathbf{M}_{\text{union}} \cup \mathbf{M}_k, \mathbf{M}_{\text{gt}})$. If $\text{IoU}^* > \text{IoU}_{\text{max}}$, we update $\pi_k \leftarrow 1$, $\mathbf{M}_{\text{union}} \leftarrow \mathbf{M}_{\text{union}} \cup \mathbf{M}_k$, and $\text{IoU}_{\text{max}} \leftarrow \text{IoU}^*$. At the end of the iterative process, an approximately optimal $\boldsymbol{\pi}^*$ can be obtained, which identifies the grid points matched to the ground truth mask. Figure 3 provides a visualization of this greedy algorithm. Finally, a mask can be represented by an unordered sequence of matched grid tokens, e.g. , `<seg><grid`$_{i_1,j_1}$`>...<grid`$_{i_n,j_n}$`><\seg>`.

### 4.2 OFFSET-AWARE DATASET CONSTRUCTION

To generate high-quality training data for offset tokens, we develop a systematic approach that categorizes grid points based on their spatial relationship to mask boundaries. Using morphological operations scaled to the offset step size, we define four distinct regions around each mask boundary (see Appendix E for detailed formulations): i) INSIDE: Stable interior points mapped to zero offset (`<OFF`$_{0,0}$`>`) ii) RING: Boundary-proximal exterior points requiring non-zero offsets iii) FAR: Distant negatives mapped to deletion (`<DELETE>`) iv) HARD-DELETE: Challenging edge cases also mapped to `<DELETE>`Each grid point is assigned to exactly one region through an ordered decision rule that prioritizes educationally valuable cases. Training pairs are sampled with bias toward INSIDE and RING regions where offset corrections provide the most learning value.

This procedure yields a variable number $K$ of grid–offset token pairs per image for supervised training. Empirically, this simulated supervision outperforms real-generated alternatives by focusing on boundary-proximal scenarios while remaining model-agnostic, creating a curated set of high-value training cases that teach effective refinement strategies. Detailed algorithms are provided in Appendix E.

## 5 REFWORDS−R1: REFWORDS FOR REINFORCEMENT LEARNING

We develop a RefWords-RL framework for referring expression segmentation (RES) and comprehension (REC) tasks, training the model to jointly output bounding boxes and masks—a format

---

[1]We use its *segment everything* mode Kirillov et al. (2023).

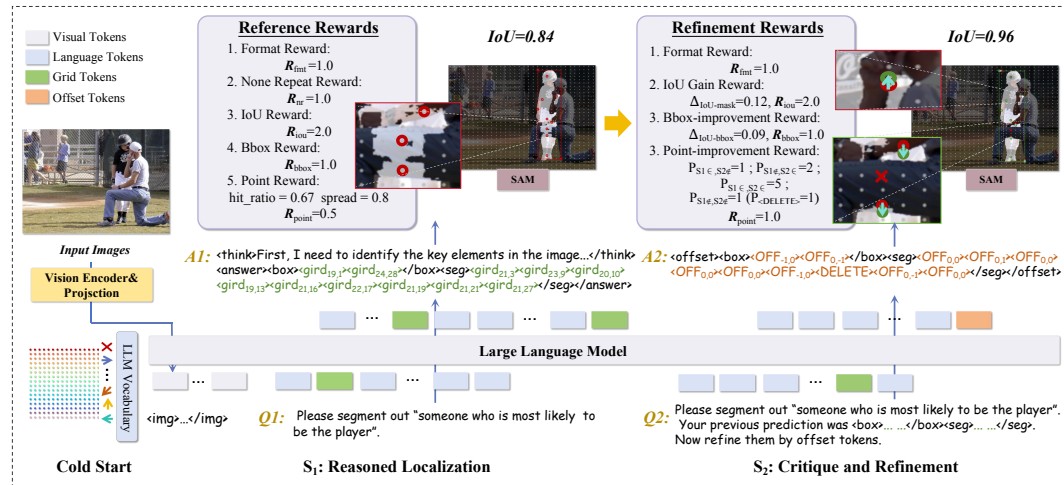

Figure 4: **Overview of RefWords-R1.** Our framework models 2D spatial localization as a two-step generative task. First, a Grid Token is generated to propose a coarse anchor region in the image. Second, an Offset Token is generated to refine this proposal to a precise point.

widely adopted in existing RL-based referring methods Liu et al. (2025a;b). Our pipeline begins with a cold-start model pre-trained via supervised fine-tuning on the RefWords vocabulary, providing the policy $\pi_\theta$ with a prior for generating spatially-grounded responses. Then, the training proceeds through a two-stage procedure using GRPO Shao et al. (2024): The first stage focuses on grid token generation with rewards for spatial accuracy and structural validity, while the second stage introduces offset tokens in multi-turn dialogues with rewards prioritizing precision improvement through iterative refinement. This self-correcting mechanism significantly enhances localization precision while maintaining conversational coherence.

### 5.1 Reward Design for Stage 1: Grid Token Generation

**Format Reward.** This reward encourages structured output with reasoning in `<think>` tags and spatial predictions in `<answer>` tags containing `<box>` and `<seg>` tokens.

**Non-repeat Reward.** This reward penalizes sentence-level repetition by returning 0 if two or more duplicate sentences appear.

**Tiered IoU Reward.** We employ SAM Kirillov et al. (2023) for automated quality assessment, converting predicted boxes and points in `<seg>` into spatial prompts to generate masks: $R_{\text{IoU}} = \sum_{k=1}^{2} \mathbb{1}(\bar{I} > t_k)$ where $t = [0.50, 0.80]$.

Given $P$ predictions and $G$ ground-truth instances matched via Hungarian assignment $\mathcal{M}$. Then, we can compute the instance-level rewards for bounding boxes and semantic-critical points as follows:

**Bboxes Reward.** We score each match based on bbox IoU and corner points distance. The reward is computed as $R_{\text{bbox}} = \frac{1}{2\max(P,G)} \sum_{(p,g)\in\mathcal{M}} \left[ \mathbb{1}\text{IoU}_{p,g} > 0.5 + \mathbb{1}|\hat{\mathbf{b}}_p - \mathbf{b}_g|_1/4 < 18 \right]$.

**Semantic-Critical Points Reward.** This reward evaluates point set quality for segmentation by combining hit ratio (fraction of points inside ground-truth masks) and spatial distribution (normalized nearest-neighbor distance). To balance point count and avoid degenerate outputs, we apply an exponential saturation term $(1 - e^{-m_p/5})$ to discourage sparse predictions and a linear penalty $(0.02m_p)$ to prevent excessive points. More reward details are list in App. F.1.

### 5.2 Reward Design for Stage 2: Offset Token Refinement

**Format Reward.** Unlike the grid stage, we found the `<think>` preamble provides negligible benefit for offsets, so the formatter enforces a minimal schema: per-instance `<offset>` tokens containing `<box>` and `<seg>` serializations.

**Point Refinement Reward.** This reward evaluates point-level refinement quality using ternary scoring $s_{k,p} \in -1, 0, 1$ per point, where $-1$ for inside→outside movements; $+1$ for outside→inside corrections, maintained inside positions, or valid deletions; $0$ otherwise. A deletion is considered valid only if the model predicts <DELETE>and no point in the $3 \times 3$ neighborhood of the original position lies inside the ground truth mask. The instance-level reward averages over its points: $R_{\text{off}}^{(k)} = \frac{1}{P_k} \sum_{p=1}^{P_k} s_{k,p}$.

**Box Refinement Reward.** This reward measures IoU gain between initial and refined bounding boxes, where $+1$ per instance if refined box IoU > initial proposal IoU, 0 otherwise.

**IoU Gain Reward.** This reward measures normalized IoU improvement between initial prediction (R1) and refined prediction (R2) predictions. Let it be $R_{\Delta\text{IoU}} = 0$ if $\Delta \leq -0.30$ or $|\Delta| < 0.01$; $R_{\Delta\text{IoU}} = 1$ if $0 < \Delta/(1 - \text{IoU}_{R1}) < 0.50$; $R_{\Delta\text{IoU}} = 2$ if $\Delta/(1 - \text{IoU}_{R1}) \geq 0.50$ or $\text{IoU}_{R1} \geq 0.80$, where $\Delta = \text{IoU}_{R2} - \text{IoU}_{R1}$.

# 6 EXPERIMENTS

## 6.1 EXPERIMENTAL SETUP

**Training Details.** We evaluate RefWords under two training paradigms. *For SFT*, we use the `ms_swift` framework with LoRA (rank=64), a batch size of 16, and a learning rate of $1 \times 10^{-6}$, training on publicly available corpora spanning image-level reasoning, referring grounding, and segmentation. *For RL*, we employ the GRPO algorithm via the `easy-r1` framework, initializing from a cold-start model trained on referring segmentation data and open-source multimodal instruction data (e.g., LLaVA-CoT-100k). Stage 1 GRPO training uses a 9K dataset containing LISA++ and referring segmentation samples, with a batch size of 16 (8 samples per step), learning rate of $1 \times 10^{-6}$, and weight decay of 0.01. Stage 2 refinement training, limited to 200 steps to prevent overfitting due to the concise nature of offset tokens. All experiments are conducted on 8× NVIDIA A800 GPUs using the DeepSpeed engine Rasley et al. (2020), with a grid size of 32 and an offset size of 64. Detailed dataset composition provided in Appendix Tab. 5.

**Benchmark Settings.** RefWords addresses a broad spectrum of visual referring tasks with a single, unified architecture. We conduct quantitative evaluations on six benchmarks: (i) *Referring Expression Comprehension* (REC), (ii) *Referring Expression Segmentation* (RES), (iii) *Reasoning Segmentation*, (iv) *Referring Captioning*, (v) *Generalized Referring Expression Segmentation* (gRES), and (vi) *Lane Polyline Detection*. We also provide (vii) a driving case study that mixes polylines (lanes), polygons (drivable area), and boxes (dynamic objects), demonstrating unified supervision in complex scenes. Comprehensive benchmark statistics are detailed the Appendix. G.

For *SFT-based paradigm*, we perform exhaustive validation across all seven settings (i)–(vii), establishing strong and consistent SFT baselines under a shared training recipe and decoding budget. (see Appendix for tasks iv–v) For *RL-based paradigm*, we focus on (i)–(iii), which reflect mainstream benchmarks for R1 paradigm referring models.

## 6.2 MAIN RESULTS: UNIFIED REFERRING WITH SFT

**Referring Expression Comprehension.** To evaluate RefWords for bounding box referring, we compare it with state-of-the-art MLLMs on REC tasks. As shown in Tab. 2, RefWords surpasses the Qwen2.5-VL baseline by +1.6% on average, demonstrating its effectiveness in spatial representation. The strong performance of RefWords-SFT-grid confirms that grid tokens alone enable high-quality localization, while the +0.4% gain of full RefWords-SFT—even on the AP@0.5 metric where improvements are subtle—highlights the consistent benefit of offset refinement. These results validate that with a suitable grid size $n \times n$, our discrete tokenization strategy outperforms continuous coordinate regression, despite inherent discretization errors.

**Referring Expression Segmentation.** We evaluate the segmentation capability of RefWords through an efficient matching strategy that associates masks with specific grid token sequences. This approach enables referring VLMs to achieve segmentation through a unified autoregressive loss, eliminating the need for task-specific architectures. As shown in Tab. 2, RefWords achieves 68.2% average IoU on Qwen2.5-VL, competitive with specialized methods while maintaining ar-

Table 1: **Referring Expression Comprehension** results (Acc@0.5) on the RefCOCO (+/g) datasets.

| Methods | refCOCO | | | refCOCO+ | | | refCOCOg | | Avg. |
|---|---|---|---|---|---|---|---|---|---|
| | Val. | Test-A | Test-B | Val. | Test-A | Test-B | Val. | Test | |
| *—— Supervised Fine-Tuning Models ——* | | | | | | | | | |
| VisonLLM Chen et al. (2023) | 87.0 | 90.6 | 80.2 | 81.6 | 87.4 | 72.1 | 82.3 | 82.2 | 82.9 |
| UNINEXT-L Yan et al. (2023) | 91.4 | 93.7 | 88.9 | 83.1 | 87.9 | 76.2 | 86.9 | **87.5** | 87.0 |
| Shikra Chen et al. (2023) | 87.0 | 90.6 | 80.2 | 81.6 | 87.4 | 72.1 | 82.3 | 82.2 | 82.9 |
| Ferret You et al. (2023) | 87.5 | 91.4 | 82.5 | 80.8 | 87.4 | 73.1 | 83.9 | 84.8 | 83.9 |
| InternVL2-8B Chen et al. (2024) | 87.1 | 91.1 | 80.7 | 79.8 | 87.9 | 71.4 | 82.7 | 82.7 | 82.9 |
| Groma Ma et al. (2024) | 89.5 | 92.1 | 86.3 | 83.9 | 88.9 | 78.1 | 86.4 | 87.0 | 86.5 |
| Qwen2.5-VL-7B Bai et al. (2025) | 90.0 | 92.5 | 85.4 | 84.2 | 89.1 | 76.9 | 87.2 | 87.2 | 86.6 |
| RefWords-SFT-grid | 90.4 | 93.8 | 86.9 | 86.3 | 90.8 | 79.4 | 87.1 | 87.5 | 87.8 |
| RefWords-SFT | **90.6** | 93.7 | **87.2** | **86.7** | **90.9** | **79.9** | **88.5** | **88.4** | **88.2** |
| *—— Reinforcement Learning Models ——* | | | | | | | | | |
| VisionReasoner[†] Wang et al. (2024) | 89.6 | 91.1 | - | 85.4 | 89.0 | - | 88.2 | **89.0** | 88.7 |
| RefWords-R1-grid | 90.2 | 92.9 | - | 86.7 | 89.9 | - | 89.2 | 88.7 | 89.6 |
| RefWords-R1 | **90.9** | **93.6** | - | **87.1** | **90.8** | - | **89.9** | 89.2 | **90.3** |

Table 2: **Referring Expression Segmentation** results on the RefCOCO (+/g) datasets.

| Methods | Training Mask Dec. | ReasonSeg | | refCOCO | | | refCOCO+ | | | refCOCOg | | Avg. |
|---|---|---|---|---|---|---|---|---|---|---|---|---|
| | | Val. | Test | Val. | T-A | T-B | Val. | T-A | T-B | Val. | Test | |
| *—— Supervised Fine-Tuning Models ——* | | | | | | | | | | | | |
| LAVT Yang et al. (2022) | ✔ | - | - | 72.7 | 75.8 | 68.8 | 62.1 | 68.4 | 55.1 | 61.2 | 62.1 | - |
| ReLA Liu et al. (2023) | ✔ | - | - | 73.8 | 76.5 | 70.2 | 66.0 | 71.0 | 57.7 | 65.0 | 66.0 | - |
| CRIS Wang et al. (2022b) | ✔ | - | - | 70.5 | 73.2 | 66.1 | 65.3 | 68.1 | 53.7 | 59.9 | 60.4 | - |
| PixelLM Ren et al. (2024) | ✔ | - | - | 73.0 | 76.5 | 68.2 | 66.3 | 71.7 | 58.3 | 69.3 | 70.5 | - |
| LISA Lai et al. (2024) | ✔ | 44.4 | 36.8 | 76.0 | 78.8 | 72.9 | 65.0 | 70.2 | 58.1 | **69.5** | 70.5 | 64.2 |
| RefWords-SFT (LLaVA-1.5) | ✘ | 44.9 | 36.9 | 74.6 | 78.4 | 71.3 | **66.4** | **72.8** | 59.8 | 68.1 | 69.8 | 64.3 |
| Qwen2.5-VL-7B Lai et al. (2024) | ✘ | 55.4 | 51.5 | 72.5 | 76.4 | 70.0 | 64.3 | 70.5 | 58.4 | 68.1 | 69.9 | 65.7 |
| RefWords-SFT-grid | ✘ | 58.1 | 54.4 | 74.3 | 77.9 | 72.3 | 65.6 | 71.9 | 58.8 | 68.0 | **70.9** | 67.2 |
| RefWords-SFT | ✘ | **59.2** | **55.8** | **76.1** | **79.2** | **73.2** | **66.4** | 72.3 | **59.9** | 69.4 | 70.9 | **68.2** |
| *—— Reinforcement Learning Models ——* | | | | | | | | | | | | |
| Seg-Zero Lai et al. (2024) | ✘ | 62.6 | 57.5 | - | 80.3 | - | - | 76.2 | - | - | 72.6 | 69.8 |
| SAM-R1 Lai et al. (2024) | ✘ | 64.0 | 60.2 | - | 79.2 | - | - | 74.7 | - | - | 73.1 | 70.2 |
| VisionReasoner Lai et al. (2024) | ✘ | 66.3 | 63.6 | - | 79.3 | - | - | 72.2 | - | - | 72.2 | 70.7 |
| RefWords-SFT-grid | ✘ | 59.6 | 60.0 | - | 79.0 | - | - | 75.9 | - | - | 72.9 | 69.5 |
| RefWords-R1 | ✘ | 64.8 | 64.0 | - | 80.9 | - | - | 77.6 | - | - | 75.1 | 72.5 |

chitectural simplicity. Offset tokens provide critical gains in RES (+1.0% IoU over grid-only), as precise positioning is essential for segmentation—minor errors in point locations are amplified during mask decoding (e.g., by SAM), significantly affecting final mask quality.

Figure 5(a) shows that by using a non-finetuned SAM, we fully preserve its generalization capability, yielding high-quality masks with fine-grained edge details. We note this can sometimes lead to discrepancies when measured against lower-quality ground-truth annotations. Figure 5 (b)(d) showcase the adaptability of our refinement mechanism, which applies small corrections to accurate proposals (b) and large corrections to less precise ones (d). Figure 5(c) specifically showcasing the effectiveness of our propose-then-refine approach on small targets, where precise localization is particularly challenging.

## 6.3 MAIN RESULTS: REFWORDS-RL

**Referring Expression Comprehension.** Our RL-based approach (RefWords-R1) achieves a notable 90.3% average accuracy, surpassing the VisionReasoner baseline by +1.6%. However, we observe that the performance gain over SFT on REC benchmarks is relatively limited. We attribute this to the inherently straightforward nature of REC tasks, which often require minimal reasoning to yield correct answers. In contrast, the integration of offset tokens demonstrates clearer advantages under the RL paradigm, as the fixed and compact search space of the $m^2 + 1$ offset tokens is better suited for RL exploration, enabling more stable and efficient policy learning compared to unconstrained coordinate regression.

**Reason and Referring Expression Segmentation.** RefWords-R1 achieves state-of-the-art performance on RES tasks (72.5% avg. IoU), demonstrating the full advantage of our token design under

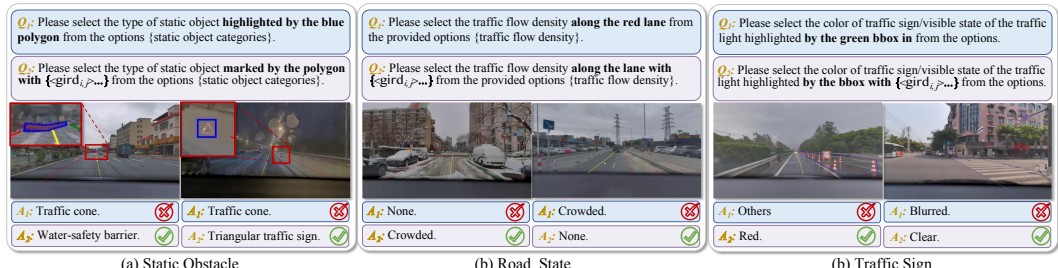

Figure 5: **RefWords-SFT Qualitative Results** We visualize the two-step localization: red dots are grid-token proposals, blue lines show the applied offset vectors, and green dots represent the final offset-refined points.

Figure 6: **Qualitative results of the proposed grid tokens in the driving scene.** Challenging examples from three referring categories demonstrate that the proposed anchor tokens offer superior region-referencing ability compared to conventional visual referring prompts.

reward-based optimization. The grid tokens provide a robust initial localization that enables efficient exploration, while the offset tokens allow the RL policy to learn precise boundary refinements that are critical for high-quality segmentation. The significant +4.3% gain over the SFT baseline underscores that the structured, finite action space of the $m^2 + 1$ offset tokens is particularly well-suited for RL, enabling stable and effective policy learning that directly translates into improved mask accuracy and generalization. Quantitative analysis reveals that the model learns effective refinement strategies: the <DELETE>is utilized in 12% of cases where initial proposals are fundamentally misplaced, preventing erroneous predictions. The offset acceptance rate serves as a reliable confidence metric, with high-acceptance predictions showing a 0.92 correlation with actual IoU improvement.

## 6.4 CASE STUDY: DRIVING SCENE APPLICATION

We further evaluate grid tokens on a proprietary driving dataset containing diverse urban scenarios with three annotation types: **lanes** (polylines), **static obstacles** (bounding boxes), and **traffic signs** (key points). For lane detection, RefWords converts continuous coordinate regression into discrete point selection, achieving +3% precision, +18% recall, and +10% F1-score over coordinate-based methods (Table 4 in Appendix C), demonstrating particular strength in handling curved lanes. For general scene understanding, RefWords consistently outperforms traditional visual prompts across all tasks (Table 3 in Appendix C), with notable gains in challenging scenarios: +12.24% for traffic sign color recognition and +7.95% for static obstacle classification. Figure 6 illustrates RefWords' precision in complex driving scenarios, validating its ability to handle diverse reference types through a unified representation without architectural modifications.

## 7 CONCLUSION

We present RefWords, a unified lexicalized representation that bridges the spatial understanding gap in multimodal learning by discretizing geometric primitives into a shared vocabulary. Through comprehensive evaluation, RefWords achieves state-of-the-art performance across diverse referring tasks under both supervised and reinforcement learning paradigms, demonstrating that structured tokenization enables precise spatial reasoning while maintaining architectural simplicity. Our work establishes a new direction for native 2D understanding in autoregressive models.

## 8 ETHICS STATEMENT

This research adheres to the ethical guidelines of the ICLR community. Our work focuses on developing spatial referring representations for multimodal large language models and does not involve collection of sensitive personal information or data that may compromise individual privacy. All datasets used in this study are publicly available referring expression benchmarks (e.g., RefCOCO, ReasonSeg) that have been released under appropriate licenses for research purposes. We carefully ensured compliance with dataset usage policies and did not perform any data manipulation that would raise ethical concerns.

Potential societal impacts of our work include both positive and negative aspects. On the positive side, our method may advance spatial reasoning capabilities in assistive technologies, robotics, and autonomous systems, potentially improving human-computer interaction. On the negative side, enhanced spatial referring abilities could potentially be misused in surveillance applications. We acknowledge these risks and emphasize that our work is intended solely for academic research and beneficial applications. No human subjects, personally identifiable information, or harmful content were involved in this study. We believe the ethical risks of this work are minimal and manageable.

## 9 REPRODUCIBILITY STATEMENT

We are committed to ensuring the reproducibility of our results, in accordance with the ICLR reproducibility guidelines. We will release the complete code implementation of RefWords, including both supervised fine-tuning and reinforcement learning frameworks, upon publication. All datasets used in our experiments are publicly available referring benchmarks.

We provide comprehensive experimental details in the paper: architecture specifications (including grid size $n \times n$, offset range $m \times m$, and token embedding dimensions), training hyperparameters (learning rates, batch sizes, optimizer configurations for both SFT and RL paradigms), and reward function formulations. Detailed descriptions of our propose-and-refine mechanism, data processing pipelines, and evaluation protocols are documented in the methodology section and appendix.

Experiments were conducted on standard NVIDIA A800 GPUs GPU. We believe these measures provide sufficient information for independent verification and reproduction of our results.

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

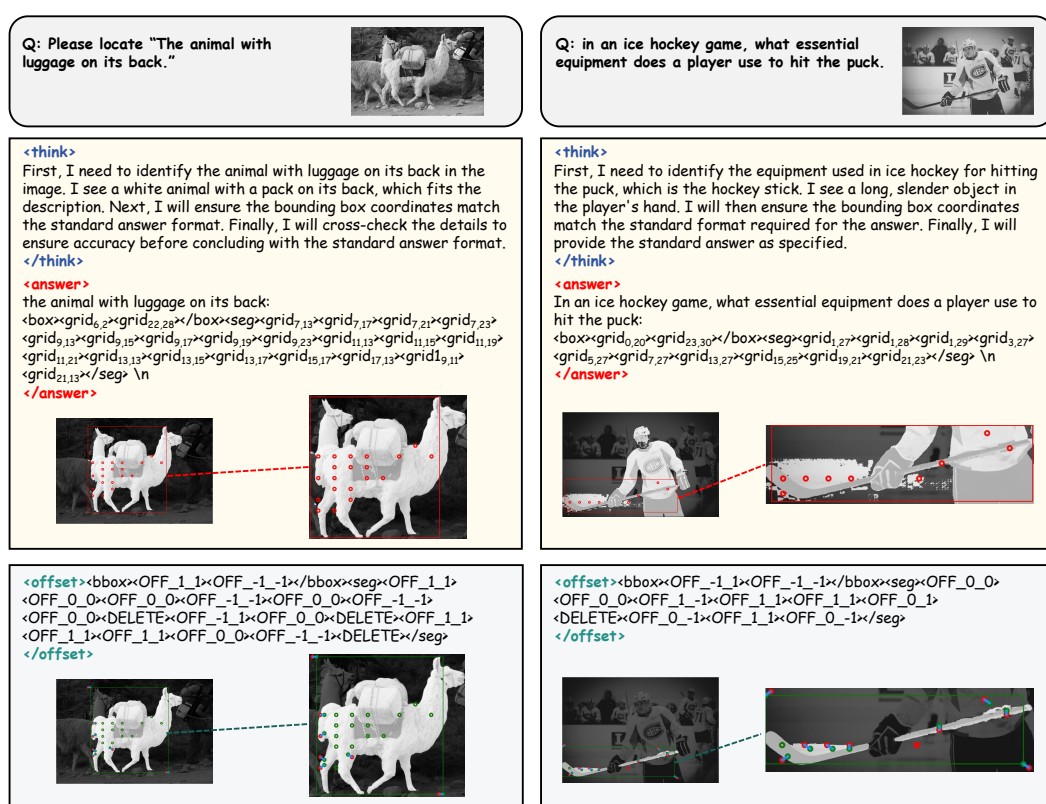

Figure 7: Additional examples demonstrating RefWords' propose-and-refine mechanism. Grid tokens establish initial spatial proposals (red dots), while offset tokens enable fine-grained adjustments through iterative refinement (blue arrows). The cases showcase how this approach handles objects of varying sizes and complexities, with particular effectiveness on small targets where precise localization is challenging.

## A  USE OF LARGE LANGUAGE MODELS

In the preparation of this manuscript, we used large language models (ChatGPT) solely for improving writing clarity and style. All scientific content—including research ideas, methodological design, experiments, analysis, and conclusions—was originated and conducted by the human authors. The models were not used to generate any scientific insights, technical implementations, or data interpretations. All LLM-generated text was carefully reviewed and modified by the authors to ensure accuracy and alignment with our original research. No confidential or unpublished data were disclosed during this process.

## B  MORE CASES FOR REFWORDS

Figure 7 presents additional qualitative examples illustrating the propose-and-refine workflow of Re-fWords. The left panel demonstrates that for points clearly inside the mask, the model refrains from unnecessary adjustments and focuses refinement efforts on boundary points. The right panel shows an extreme case of a slender mask where RefWords still achieves accurate proposal generation and precise correction. These cases demonstrate the versatility of our approach across different spatial referring scenarios, highlighting its ability to maintain precise localization through the coordinated operation of grid and offset tokens. The visualizations underscore how RefWords achieves robust spatial understanding while operating within standard autoregressive frameworks.

## C  DRIVING SCENE TESTING

### C.1  DATASET ANNOTATIONS

We constructed a proprietary autonomous driving dataset to validate our Grid Tokens in complex scenarios in fair comparison with state-of-the-art approaches. This dataset contains 1,988 training samples (29,825 annotations) and 980 test samples (14,524 annotations), covering diverse urban scenarios like intersections, highways, and pedestrian zones. As illustrated in Figure 8(a), the dataset categorizes driving targets into three classes: Traffic Lanes, Static Obstacles, and Traffic Signs with hierarchical annotations for multi-granular reasoning. We then design a series of classification tasks to evaluate the model's ability to understand and refer to these specific regions. For example, the model is designed to assess whether a designated lane is obstructed and to identify the type of obstruction (e.g., construction or a dynamic object such as a vehicle that is stationary or moving slowly). Common static objects in driving scenes include, but are not limited to, traffic cones, parking poles, and warning signs. This capability enables the model to provide detailed descriptions of static obstacles and their potential impact on driving safety. For traffic signs or lights, the model can be trained to identify potential safety hazards, such as occlusions or damage. This allows the model to assess whether a traffic sign is fully visible or partially obscured, which is critical to ensuring safe navigation for autonomous driving. The color classification task is set to help determine the sign type. Figure 8(b) presents a sample from our dataset. For each sample, we classify the object categories presented in the image according to the selection options depicted in Figure 8(a).

### C.2  RESULTS AND ANALYSIS

In Section 6.4 of the main paper, we present both qualitative and quantitative experimental results. The findings show that grid tokens yield a significant improvement over traditional visual prompt-based referring. Visualizations further reveal that grid tokens are particularly effective at localizing small, distant objects, as their discrete representation aids the model in precisely understanding and pinpointing these challenging targets.

Our experiments further demonstrate that integrating Chain-of-Thought (CoT) reasoning significantly enhances static object classification in driving scenarios. By providing detailed descriptions for each object category and guiding the model to analyze the visual features of the referring target, we achieve more accurate classification. For example, we define a set of static object categories along with their visual characteristics (e.g., Traffic Cone: conical structure; Traffic Warning Pole: slender cylindrical shape; Diversion Sign: features directional arrows). The task prompt instructs the model to identify the color, texture, and shape of the object(s) marked by the grid tokens and then select the correct category from the provided list. This CoT-guided approach significantly improves the model's ability to localize and classify static objects accurately in complex driving scenes.

Table 3: Performance comparison using Ref-Words in driving scenes.

| Methods | Lane Polyline | | |
|---|---|---|---|
| | Precision | Recall | F1 |
| Coords-based | 0.49 | 0.47 | 0.48 |
| Grid Tokens | 0.52 | 0.65 | 0.58 |

Table 4: Polyline grounding results in driving scenes.

| Category | Task | Baseline | Grid Tokens |
|---|---|---|---|
| Static obstacle | Classification | 81.69 | 89.64 |
| | Visible State | 90.60 | 93.49 |
| Road | Blockage Status | 86.07 | 87.25 |
| | Surface Condition | 95.46 | 95.68 |
| | Traffic Density | 84.31 | 86.39 |
| Traffic Sign | Color | 71.43 | 83.67 |
| | Visible State | 63.27 | 67.35 |

## D  COORDINATE TRANSFORMATION DETAILS

**Grid Token Coordinate Mapping.** Given an image $\mathbf{I} \in \mathbb{R}^{H \times W \times 3}$ and the normalized plane $\Omega = [0,1]^2$ with $(u,v) = (x/W,\ y/H)$, we tile $\Omega$ with an $n \times n$ grid whose cell centers are:

$$\mathbf{c}_{i,j} = (x_{i,j}, y_{i,j})^\top = \left( \tfrac{j+0.5}{n}\, W,\ \tfrac{i+0.5}{n}\, H \right)^\top, \qquad i,j \in \{0,\dots,n-1\}. \tag{2}$$

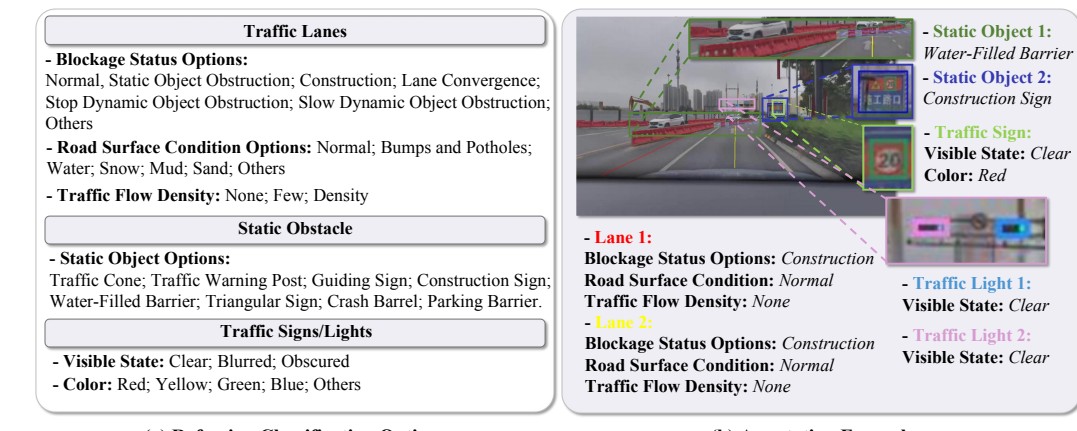

(a) Referring Classification Options  (b) Annotation Example

Figure 8: **Overview of driving dataset annotations information.**

We introduce $n^2$ learnable grid tokens $\{\texttt{<grid}_{i,j}\texttt{>}\}_{i,j=0}^{n-1}$ and extend the base vocabulary from $\mathcal{V}_{\text{LLM}}$ to $\mathcal{V}_{\text{LLM}} \cup \{\texttt{<grid}_{i,j}\texttt{>}\}$, binding each $\texttt{<grid}_{i,j}\texttt{>}$ to $\mathbf{c}_{i,j}$. A point $(x, y)$ is mapped to the nearest cell via:

$$(i, j) = \arg \min_{(i',j') \in \{0,\ldots,n-1\}^2} \left\| (x, y) - \mathbf{c}_{i',j'} \right\|_2, \tag{3}$$

and is then tokenized as $\texttt{<grid}_{i,j}\texttt{>}$.

**Offset Token Refinement Process.** We define the offset token set as:

$$\mathcal{T}_{\text{offset}} = \{\texttt{<OFF}_{\delta_u,\delta_v}\texttt{>} \mid \delta_u, \delta_v \in \{-1, 0, 1\}\} \cup \{\texttt{<DELETE>}\}.$$

Let $m$ be the global offset granularity and define step sizes:

$$s_x = W/m, \qquad s_y = H/m, \qquad \mathbf{S} = \text{diag}(s_x, s_y). \tag{4}$$

Given an grid $\texttt{<grid}_{i,j}\texttt{>}$ with center $\mathbf{c}_{i,j}$, the model emits either $\texttt{<OFF}_{\delta_u,\delta_v}\texttt{>}$ or $\texttt{<DELETE>}$. Let $\boldsymbol{\delta} = (\delta_u, \delta_v)^\top$; applying an offset yields the refined coordinate:

$$\mathbf{p}' = \mathbf{c}_{i,j} + \mathbf{S}\,\boldsymbol{\delta}, \qquad \boldsymbol{\delta} \in \{-1, 0, 1\}^2, \tag{5}$$

optionally followed by clipping $(x', y')$ to $[0, W] \times [0, H]$.

The complete vocabulary is defined as:

$$\mathcal{V} = \mathcal{V}_{\text{LLM}} \cup \mathcal{T}_{\text{grid}} \cup \mathcal{T}_{\text{offset}}. \tag{6}$$

# E  OFFSET-AWARE DATASET CONSTRUCTION DETAILS

## E.1  MORPHOLOGICAL OPERATIONS AND REGION DEFINITIONS

Let $\mathbf{M}_{\text{gt}} \in \{0, 1\}^{H \times W}$ be the binary foreground mask. We place an $n \times n$ anchor grid and denote the pixel center of cell $(i, j)$ by $\mathbf{c}_{i,j} = (x_{i,j}, y_{i,j})^\top$. The offset step sizes $s_x, s_y$ and scaling matrix $\mathbf{S}$ are defined in equation equation 4 of the main text.

To construct pools of candidate grid tokens, we employ morphology-based bands scaled according to the offset step size. Let $\mathcal{K}_k \in \{0, 1\}^{k \times k}$ represent a square structuring element with side length $k$ pixels. We define:

$$\begin{aligned} k_e = \lfloor s_y \rfloor + 1, \quad & \mathbf{E} = \mathbf{M}_{\text{gt}} \ominus \mathcal{K}_{k_e}, \\ k_d = 2\lfloor s_y \rfloor + 1, \quad & \mathbf{D} = \mathbf{M}_{\text{gt}} \oplus \mathcal{K}_{k_d}, \end{aligned} \tag{7}$$

where $\lfloor \cdot \rfloor$ denotes the floor operation, while $\ominus$ and $\oplus$ represent morphological erosion and dilation respectively.

---

**Algorithm 1:** Constructing Offset-Supervised Conversations

---

**Input:** Referring dataset $\mathcal{D}$; grid size $n$; offset granularity $m$; IoU threshold $\tau$
**Output:** JSONL conversations containing grids and offset targets
**foreach** $(I, \mathbf{M}_{gt}, q) \in \mathcal{D}$ **do**

    Resize $I, \mathbf{M}_{gt}$ to $H \times W$; compute $s_x = W/m$, $s_y = H/m$, $\mathbf{S} = \mathrm{diag}(s_x, s_y)$;
    // grid pools via morphology (cf. equation 7--equation 8)
    Compute $\mathbf{E}, \mathbf{D}, \mathbf{B}$; assign each grid cell $(i, j)$ to one of INSIDE/RING/FAR/HARD-DELETE by rule
     equation 10;
    // Segmentation grids and offsets
    Sample $K$ grids $\{(i_k, j_k)\}_{k=1}^K$ from the pools;
    **for** $k = 1$ **to** $K$ **do**
        **if** $\mathbf{M}_{gt}(y_{i_k}, x_{i_k}) = 1$ **then**
            emit [OFF_0_0]
        **else if** $\mathrm{Hit}_{3 \times 3}(i_k, j_k)$ **then**
            pick $(\delta_u, \delta_v) \in \{-1, 0, 1\}^2$ with $\mathbf{M}_{gt}\big(\mathrm{clip}(\mathbf{c}_{i_k, j_k} + \mathbf{S}\boldsymbol{\delta}; [0, W-1] \times [0, H-1])\big) = 1$ and
             emit [OFF_$\delta_u$_$\delta_v$]
        **else**
            emit <DELETE>

    // Bounding-box corner offsets
    Let $B^\star \leftarrow \mathrm{BBox}(\mathbf{M}_{gt})$; jitter its TL/BR to grid corners $(i_{tl}, j_{tl}), (i_{br}, j_{br})$;
    Evaluate all 81 offset pairs for the two corners (apply $\mathbf{S}$-scaled displacements), obtain $\mathrm{IoU}_{max}$;
    **if** $\mathrm{IoU}_{max} \geq \tau$ **then**
         emit the two corner offsets
    **else**
         emit <DELETE> for both corners
    // Serialization
    Write a JSONL sample with image tag, user prompt $q$ and grids (user turn), and the offsets (assistant
     turn);

---

A thin boundary band is additionally defined as:

$$\mathbf{B} = \mathrm{dilate}(\mathrm{grad}(\mathbf{M}_{gt}), \mathcal{K}_b), \tag{8}$$

where $\mathrm{grad}(\mathbf{M}_{gt})$ is the morphological gradient and $b$ is a small width parameter.

By construction, $\mathbf{E} \subset \mathbf{M}_{gt} \subset \mathbf{D}$: $\mathbf{E}$ forms a step-sized interior buffer, $\mathbf{D}$ creates a step-sized exterior halo, and $\mathbf{B}$ captures edge uncertainty as a narrow boundary ribbon.

### E.2   GRID POINT CATEGORIZATION AND SAMPLING

We define a one-step hit test to determine reachability:

$$\mathrm{Hit}(i, j) \triangleq \exists \boldsymbol{\delta} \in \{-1, 0, 1\}^2 : \mathbf{M}_{gt}(\mathbf{c}_{i,j} + \mathbf{S}\boldsymbol{\delta}) = 1. \tag{9}$$

Each grid center is assigned to exactly one category via the hierarchical decision rule:

$$\mathrm{pool}(i, j) = \begin{cases} \text{HARD-DELETE}, & \mathbf{B}(y_{i,j}, x_{i,j}) = 1 \wedge \mathbf{M}_{gt}(y_{i,j}, x_{i,j}) = 0 \wedge \neg \mathrm{Hit}(i, j), \\ \text{INSIDE}, & \mathbf{E}(y_{i,j}, x_{i,j}) = 1, \\ \text{RING}, & \mathbf{D}(y_{i,j}, x_{i,j}) = 1 \wedge \mathbf{M}_{gt}(y_{i,j}, x_{i,j}) = 0, \\ \text{FAR}, & \text{otherwise.} \end{cases} \tag{10}$$

Following pool formation $\mathcal{P}_{hard} \rightarrow \mathcal{P}_{inside} \rightarrow \mathcal{P}_{ring} \rightarrow \mathcal{P}_{far}$, we sample $K \sim \pi_K$ grids per image with preferential selection from $\mathcal{P}_{inside}$ and $\mathcal{P}_{ring}$, while maintaining representation from all categories for robustness.

We systematically compare two supervision strategies: (i) simulated pairs, constructed directly from ground-truth masks using the one-step offset test, and (ii) real-generated pairs, obtained by first inferring grid tokens with an internal annotator and then deriving offsets. Empirical results show that simulated supervision consistently outperforms the real-generated variant. Crucially, it remains annotator-agnostic, concentrating samples on boundary-proximal and near-miss scenarios—resulting in a curated set of informative hard cases while mitigating bias from generator failures.

## F  REWARD DETAILS

### F.1  GRID TOKENS INSTANCE-LEVEL REWARD

From each non-empty line in `<answer>` we extract at most one predicted instance $p$ consisting of an optional box $\hat{\boldsymbol{b}}_p \in \mathbb{R}^4$ and a point set $\mathcal{P}_p = \{\boldsymbol{q}\} \subset \mathbb{R}^2$. Let there be $P$ predictions and $G$ ground-truth (GT) instances with binary masks $\{M_g\}_{g=1}^G$ and tight boxes $\{b_g\}_{g=1}^G$. We define pairwise similarities between predicted $p$ and ground-truth $g$:

1. Box IoU:

$$\mathrm{IoU}_{p,g} \in [0,1]. \tag{11}$$

2. Point-hit ratio: the fraction of predicted points that land inside $\mathbf{M}_{\mathrm{gt}}$,

$$H_{p,g} = \frac{1}{\max(1, |\mathcal{P}_p|)} \sum_{\mathbf{q} \in \mathcal{P}_p} \mathbb{1}\{\mathbf{q} \in \mathbf{M}_{\mathrm{gt}}\} \in [0,1]. \tag{12}$$

3. Normalized $L_1$ box score (optional):

$$S_{p,g}^{\ell_1} = \mathrm{clip}\left(1 - \frac{\|\hat{\mathbf{b}}_p - \mathbf{b}_g\|_1/4}{\tau_{\ell_1}}, \ 0, \ 1\right). \tag{13}$$

These are combined into a similarity used only for assignment:

$$\mathrm{Sim}_{p,g} = \mathrm{IoU}_{p,g} + H_{p,g} + S_{p,g}^{\ell_1}, \tag{14}$$

We solve a Hungarian assignment with costs $C_{p,g} = 3 - \mathrm{Sim}_{p,g}$, yielding matched pairs $\mathcal{M} \subseteq \{1..P\} \times \{1..G\}$. Denote $M = |\mathcal{M}|$. We use $\tau_{\ell_1} = 18$ px.

**A. Bounding-box quality** $B \in [0,1]$. For each $(p,g) \in \mathcal{M}$, define a pair score

$$\beta_{p,g} \triangleq \mathbb{1}\{\mathrm{IoU}_{p,g} > \tau_{\mathrm{IoU}}\} + \mathbb{1}\{\tfrac{1}{4}\|\hat{\mathbf{b}}_p - \mathbf{b}_g\|_1 < \tau_{\ell_1}\} \in \{0,1,2\}, \tag{15}$$

where $\tau_{\mathrm{IoU}} = 0.5$. Normalize by the larger set size:

$$B \triangleq \frac{1}{2\max(P,G)} \sum_{(p,g) \in \mathcal{M}} \beta_{p,g}. \tag{16}$$

**B. Key-Point Quality** $T \in [0,1]$. For each match $(p,g) \in \mathcal{M}$, we compute a key points quality

$$F_{p,g} \triangleq S(m_p)\Big(w_H H_{p,g} + w_{\mathrm{spr}} \mathrm{Spread}_{p,g}\Big) - \lambda_m m_p. \tag{17}$$

where $H_{p,g}$ is the hit ratio, and $\mathrm{Spread}_{p,g}$ rewards larger nearest-neighbor spacing normalized by object scale:

$$\bar{d}_p = \frac{1}{m_p} \sum_{i=1}^{m_p} \min_{j \neq i} \|\boldsymbol{q}_i - \boldsymbol{q}_j\|_2, \qquad \mathrm{Spread}_{p,g} = \mathrm{clip}\big(\bar{d}_p/(\rho_s r_g), \ 0, \ 1\big). \tag{18}$$

The multiplicative saturation $S(m) = 1 - \exp(-m/m_0)$ discourages degenerate few-point outputs, and the linear term $\lambda_m m_p$ penalizes overly long point lists. We aggregate across matches with point-count weighting:

$$T = \mathrm{clip}\left(\frac{\sum_{(p,g) \in \mathcal{M}} m_p F_{p,g}}{\sum_{p=1}^P \max(1, m_p)}, \ 0, \ 1\right). \tag{19}$$

We set $w_H = 0.6$, $w_{\mathrm{spr}} = 0.4$, $\lambda_m = 0.02$, $\rho_s = 0.30$.

Table 5: Summary of training data composition.

| Training | Composition |
|---|---|
| **SFT Training** | Image reasoning (LLaVA-665K, 665K), referring grounding (RefCOCO/+/g, 65.8K), segmentation (COCO-Stuff+ADE20K, 164K), captioning (Visual Genome, 108K), part-level segmentation (PACO-LVIS+PASCAL-Part, 85.9K), multi-instance segmentation (gRefCOCO, 49.8K) |
| **Cold Start** | Referring datasets (RefCOCO/+/g, 65.8K), offset training (RefWords-Off, 56.3K), instruction tuning (LLaVA-CoT-100K, 100K) |
| **GRPO Training** | Single-target referring (RefCOCOg subset, 3K) + multi-instance cases (LISA++ and gRefCOCO, 6.0K) |

# G    MORE TRAINING DETAILS

We perform quantitative evaluations on the following tasks involving visual referring:

**(i) Referring Expression Comprehension (REC)** assesses the model's ability to localize textual descriptions by predicting bounding boxes for referred objects, demonstrating grid Tokens' superiority in bounding box grounding over coordinate-based methods.

**(ii) Referring Captioning** evaluates region understanding given referring inputs (e.g., bbox, mask), showing that grid Tokens enable precise region referencing without the need for separately designed region-specific architectures.

**(iii) Referring Expression Segmentation (RES)** extends localization to pixel-level mask prediction, illustrating grid Tokens' capability to handle complex mask representations without task-specific design and auxiliary losses.

**(vi) Generalized RES** validates multi-instance resolution through grid sequences, supporting simultaneous references to multiple objects.

**(v) Lane Polyline Detection** demonstrates that ordered grid token sequences outperform coordinate strings in representing topological structures (e.g., curved lanes).

Additionally, we deploy RefWords in driving scenarios, showcasing their practical advantages in resolving real-world queries.

# H    MORE TRAINING RESULTS

**Referring Captioning.** We assess region-based caption generation on refCOCOg Mao et al. (2016) and Visual Genome Krishna et al. (2017). RefWords achieves performance comparable to models utilizing specialized region feature extractors (✓), confirming the efficacy of RefWords in enhancing region-aware comprehension. RefWords excels in handling scenarios with overlapping objects, where traditional bounding boxes often fail to capture targeted regions precisely.

Table 6: **Region-Level Captioning** results on the refcocog and visual genome datasets.

| Model | Region Feat. Extractor | refCOCOg | | Visual Genome | |
|---|---|---|---|---|---|
| | | METEOR | CIDEr | METEOR | CIDEr |
| GRIT Wu et al. (2024) | ✔ | 15.2 | 71.6 | 17.1 | 142.0 |
| SLR Yu et al. (2017) | ✔ | 15.9 | 66.2 | - | - |
| GPT4RoI Zhang et al. (2023) | ✔ | - | - | 17.4 | 145.2 |
| GLaMM Rasheed et al. (2024) | ✔ | 16.2 | 106.0 | 19.7 | 180.5 |
| Groma Ma et al. (2024) | ✔ | 16.8 | 107.3 | 19.0 | 158.4 |
| Kosmos-2 Peng et al. (2023) | ✘ | 14.1 | 62.3 | - | - |
| **Ours** | ✘ | 14.7 | 107.4 | 17.9 | 153.2 |

**Generalized RES.** RefWords naturally support multi-instance expressions. We validate the effectiveness of our method for multi-instance segmentation on the gRefCOCO dataset. The results on

| Methods | Training M-Dec. | refCOCO | | | refCOCO+ | | | refCOCOg | | Avg. |
|---|---|---|---|---|---|---|---|---|---|---|
| | | Val. | T-A | T-B | Val. | T-A | T-B | Val. | Test | |
| MCN Luo et al. (2020) | ✔ | 62.4 | 64.2 | 59.7 | 50.6 | 55.0 | 44.7 | 49.2 | 49.4 | 54.4 |
| VLT Ding et al. (2021) | ✔ | 67.5 | 70.5 | 65.2 | 56.3 | 61.0 | 50.1 | 55.0 | 57.7 | 60.4 |
| LAVT Yang et al. (2022) | ✔ | 72.7 | 75.8 | 68.8 | 62.1 | 68.4 | 55.1 | 61.2 | 62.1 | 65.8 |
| CRIS Wang et al. (2022b) | ✔ | 70.5 | 73.2 | 66.1 | 65.3 | 68.1 | 53.7 | 59.9 | 60.4 | 64.7 |
| PixelLM Ren et al. (2024) | ✔ | 73.0 | 76.5 | 68.2 | 66.3 | 71.7 | 58.3 | 69.3 | 70.5 | 69.2 |
| LISA Lai et al. (2024) | ✔ | 76.0 | 78.8 | 72.9 | 65.0 | 70.2 | 58.1 | 69.5 | 70.5 | 70.1 |
| **Ours** | ✘ | **74.6** | **78.4** | **71.3** | **66.4** | **72.8** | **59.8** | **68.1** | **69.8** | **70.2** |

Table 7: **Generalized Referring Expression Segmentation** results (cIoU) on the RefCOCO (+/g) datasets.

the gRefCOCO demonstrate the effectiveness of RefWords in multi-instance segmentation, achieving competitive performance compared to specialized methods while maintaining architectural simplicity.

# I ABLATION AND ANALYSIS

## I.1 GRID RESOLUTION

For bounding box–based referring, the length is fixed to two corner points, but in segmentation tasks, the grid size $n$ must balance spatial precision and prediction complexity. Although a larger $n$ yields finer granularity, it also increases the number of grid tokens per mask. We investigate how $n$ affects segmentation performance under this trade-off.

| Grid Size | gIoU | cIoU | Average Token Length |
|---|---|---|---|
| $32 \times 32$ | 68.2 | 65.8 | 8.7 |
| $16 \times 16$ | 68.9 | 66.2 | 5.2 |

Table 8: **Performance comparison of different grid resolution.**

As Table 8 shows, increasing the grid resolution from 16×16 to 32×32 paradoxically reduces gIoU by 0.7% despite providing finer spatial granularity for referring expression segmentation. We identify two key factors contributing to this observation:

First, increasing the grid size results in more grid tokens per mask instance. On RefCOCO, the average number of grid tokens per mask increases from 5.2 to 8.7. This expanded token sequence complicates the model's task of accurately interpreting supervision signals, as longer sequences introduce more opportunities for segmentation errors.

Second, the extended output sequence heightens prediction error susceptibility. For SAM-based segmentation, errors in point token prompts have amplified consequences—even minor token misplacements can significantly degrade mask quality. While finer granularity appears beneficial theoretically, the practical trade-off involves increased complexity and error propagation that ultimately impairs performance in precision-sensitive segmentation tasks.

To maintain consistency with bounding box tasks while avoiding redundancy, we adopt a two-stage approach: we first generate ground-truth annotations using SAM with n=16 grid resolution, then map these 16×16 points to the 32×32 grid space for model training. This strategy preserves the benefits of fine-grained spatial representation while mitigating the error accumulation associated with direct high-resolution prediction.

Our future work will consider using Hybrid approaches (e.g., adaptive grids) to resolve this problem.

## I.2 IMAGE PREPROCESSING

We evaluate the impact of three common image preprocessing strategies—center cropping, resizing, and padding—on localization performance. Among them, padding yielded the worst performance. This is because padding introduces a gray border around the image, which reduces the resolution of the valid content, negatively affecting the model's ability to focus on relevant features. In contrast,

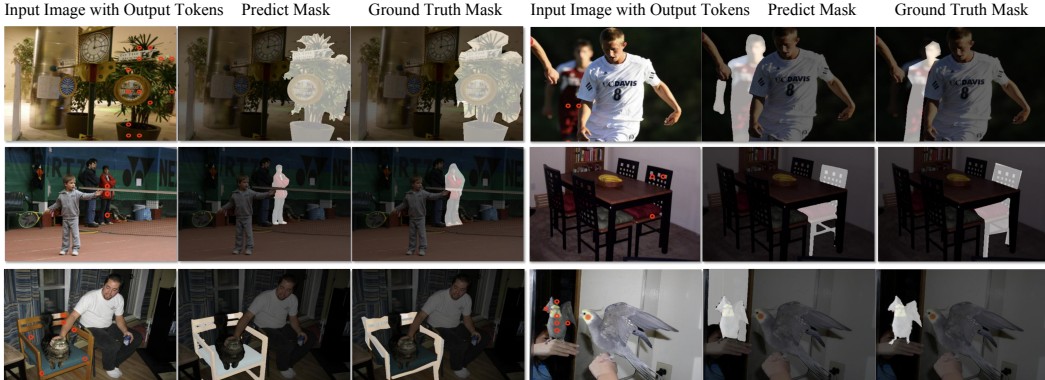

| Input Image with Output Tokens | Predict Mask | Ground Truth Mask | Input Image with Output Tokens | Predict Mask | Ground Truth Mask |

Figure 9: **More qualitative results of the segmentation task.** From top to bottom, the predictions are ordered by decreasing Intersection-over-Union (IoU) scores relative to the ground truth masks.

center crop can alter the semantic content of the image. For example, in the task of identifying "the person on the far left," if the leftmost person is cropped out, the ground truth will no longer align with the cropped image, leading to mismatched predictions. On the other hand, resizing the image consistently provided the best results, maintaining the integrity of the image content while scaling it to a uniform size suitable for processing by the model. Future work may explore any resolution strategies to further enhance performance.

## J   ADDITIONAL QUALITATIVE RESULTS AND ANALYSIS

Figure 9 presents additional qualitative results showcasing the output grid tokens, prediction masks, and their corresponding ground truth (GT) masks. The results are organized from top to bottom, ranging from predictions that are more precise than the GT mask to some failure cases. These visualizations highlight the following key observations:

(1) High-Quality Predictions: The model is capable of generating highly accurate grid tokens, which align well with the GT masks. These results demonstrate the effectiveness of grid tokens in precisely localizing and referring to objects in complex scenes.

(2) Failure Cases: The failure cases reveal that, although the model can predict grid tokens with considerable accuracy, discrepancies between the grid token outputs and the mask generation by SAM can lead to segmentation errors. This observation provides insight into why segmentation scores remain lower compared to segmentation-specific methods. However, it is important to note that segmentation itself is not a necessary form of expression for referring tasks.

For real-world applications, precise segmentation is not always required for effective referring. The ability of the model to accurately predict grid tokens is often sufficient for tasks such as object localization and referring expression comprehension. The qualitative results underscore the robustness of grid tokens as a referring representation, even in cases where segmentation performance is suboptimal.

