# OpenReview forum: "Tokens that Know Where: Self-improving 2D Spatial Vocabulary for Multi-modal Understanding"
_ICLR.cc/2026/Conference — ICLR 2026 Conference Withdrawn Submission_

### Official Review · Reviewer_gA4p · 2025-10-28

**Soundness:** 3
**Presentation:** 3
**Contribution:** 2
**Rating:** 4
**Confidence:** 4

**Summary:**

This paper proposes RefWords, a learnable 2D spatial vocabulary for MLLMs that enables native spatial reasoning within a standard autoregressive architecture. The vocabulary has two parts: 1) grid tokens and 2) offset tokens. The approach supports multiple formats with a propose-then-refine decoding scheme. Experiments on REC/RES and a driving case study report consistent gains over strong MLLM baselines; offset tokens particularly help in segmentation and RL.

**Strengths:**

1. Turning 2D localization into lexicalized tokens integrates seamlessly into seq2seq MLLMs without extra heads or decoders.
2. One vocabulary handles points, boxes, masks, and polylines, reducing architectural sprawl and enabling multitask training/evaluation.
3. The offset stage operationalizes critique-and-correct behavior with discrete actions that are well-suited to RL exploration; measured gains on RES corroborate this.

**Weaknesses:**

1. For masks, decoding and some rewards use SAM from grid/point prompts. This creates a tool-in-the-loop coupling that can hide true model errors or inherit SAM failure modes.
2. An n^2 grid inflates the vocabulary. Show ablations on n (and offset granularity m) vs. accuracy, latency, and decoding length; discuss memory/compute footprint for high resolution images.
3. Fundamentally, RefWords leverages richer supervisory signals. Prior work (e.g., methods built on LISA) can achieve strong performance on segmentation—an extremely fine-grained, spatially aware task—even without such additional supervision.

**Questions:**

Does requiring the model to output grid coordinates risk challenging an LLM’s capabilities and thereby weakening its generality? It seems nontrivial for an LLM to reliably learn to predict grid indices correctly.

---

### Official Review · Reviewer_okBg · 2025-10-29

**Soundness:** 2
**Presentation:** 2
**Contribution:** 2
**Rating:** 2
**Confidence:** 4

**Summary:**

This paper addresses the challenge of spatial reasoning in multimodal large language models (MLLMs), which often lose 2D spatial information during token serialization. The authors propose RefWords, a spatial representation framework that introduces a learnable vocabulary composed of two components: Grid Tokens (for structured spatial anchoring) and Offset Tokens (for fine-grained localization refinement). By embedding spatial relationships directly in the token space, RefWords enables autoregressive MLLMs to perform 2D reasoning without architectural modifications. Experiments demonstrate consistent performance gains on referring tasks under both supervised and reinforcement learning settings, suggesting that sequential tokens can indeed learn continuous spatial mappings.

**Strengths:**

* Novel spatial representation concept: The paper introduces the notion of RefWords with a clear grid + offset design, effectively linking sequential tokens with 2D spatial positions.

* Comprehensive design and implementation: The method is well-formalized, with detailed definitions, training setup, and reward functions for reinforcement learning.

* Empirical results: RefWords achieves notable improvements on referring expression comprehension (REC) and related localization benchmarks.

* Visualization and qualitative analysis: Figures (Fig. 1 and Fig. 6) are intuitive and visually appealing, showing promising results in complex scenes such as driving environments.

**Weaknesses:**

1. **Incomplete and inaccurate citations:**
The introduction and related work sections lack sufficient references and contain citation errors (e.g., repeated or incorrect references in Table 2, lines 405–407). Formatting and typographical issues are also frequent.

2. **Weak related work review:**
The survey of multimodal referential understanding methods is partial and biased.

    * For coordinate-based models, representative works such as Qwen-VL [1] are missing, and GPT4RoI is incorrectly categorized (it relies on local features rather than coordinates).
    * For models using discrete spatial tokens, key works like Florence-2 [2], Kosmos-2, and VisionLLM [3] already perform grid-based visual grounding with specialized vocabularies.
    * More recent efforts, such as ClawMachine [4], extend this paradigm with discrete visual token reasoning.
In short, numerous works have been proposed for auto-regressively predict discrete spatial-aware tokens for visual grounding, making the proposed approach only an incremental step rather than a fundamentally new idea.

3. **Unclear base model configuration:**
The paper only vaguely states that the baseline is “Qwen-2.5-VL” without specifying whether this model was used for initialization, finetuning, or comparison. The visual encoder and training details are also omitted, hindering reproducibility.

4. **Marginal reinforcement learning benefits:**
The reported improvements from reinforcement learning are minimal and insufficiently analyzed. While Offset Tokens are introduced to refine localization, the evaluation only reports Acc@IoU = 0.5, with no quantitative evidence of IoU gains. The practical necessity and broader applicability of these tokens remain unclear.

**References**

[1] Jinze Bai et al. Qwen-VL: A Versatile Vision-Language Model for Understanding, Localization, Text Reading, and Beyond.

[2] Bin Xiao et al. Florence-2: Advancing a Unified Representation for a Variety of Vision Tasks. CVPR 2024.

[3] Wenhai Wang et al. VisionLLM: Large Language Model is also an Open-Ended Decoder for Vision-Centric Tasks.

[4] Tianren Ma et al. ClawMachine: Learning to Fetch Visual Tokens for Referential Comprehension. ICLR 2025.

**Questions:**

1. What exact base model and visual encoder were used for training RefWords? How was “Qwen-2.5-VL” integrated or modified?

2. Can the authors provide more detailed IoU-based results (e.g., IoU > 0.7) to demonstrate fine-grained localization improvements?

3. Are there qualitative examples where Offset Tokens specifically correct spatial predictions compared to the baseline?

---

### Official Review · Reviewer_56cX · 2025-10-29

**Soundness:** 3
**Presentation:** 3
**Contribution:** 2
**Rating:** 6
**Confidence:** 4

**Summary:**

The paperproposes RefWords, a simple yet general spatial token vocabulary that enablesMLLM to represent and refine 2D locations through text tokens. Instead of relying on extra prediction heads, the model uses grid tokens to discretize an image into spatial anchors and offset tokens to iteratively refine or remove anchors, forming a self-correcting propose–refine loop.

The method is trained via supervised fine-tuning and reinforcement learning. During inference, masks are generated by prompting SAM with the predicted tokens rather than using a learned decoder. Experiments on referring expression comprehension and segmentation, reasoning segmentation, polyline grounding, and a driving case study show that RefWords improves over strong MLLM baselines like Qwen2.5-VL, achieving higher accuracy and IoU, particularly when RL and offsets are used.

**Strengths:**

1. The paper is well written, and the figures are clearly designed and informative.

2. The proposed method is insightful and introduces an interesting perspective on spatial token representation.

3. The proposed approach effectively enhances model performance across multiple visual grounding tasks.

**Weaknesses:**

1. The introduction states that coordinates like “199” and “200” may be spatially close but distant in token space due to textual encoding differences. While the example is intuitive, it lacks citation or evidence. Are there existing studies or analyses that empirically show this tokenization issue or its impact on model performance?

2. Would be nice if the author can specify their based model /architecture in the experiment table alone with the model size etc to show a clear picture on the improvement.

3. Would be nice if the result also contain other advance model like LLaVA-OV, other advance Qwen family.

**Questions:**

Please address the weakness mentioned above.

---

### Official Review · Reviewer_DxhB · 2025-11-01

**Soundness:** 3
**Presentation:** 3
**Contribution:** 2
**Rating:** 4
**Confidence:** 4

**Summary:**

This paper proposes RefWords, a learnable spatial vocabulary for MLLMs consisting of Grid Tokens (dividing images into n×n anchors) and Offset Tokens. The method trains MLLMs to generate these spatial tokens for various referring tasks within a standard autoregressive framework, without requiring specialized architectural components. Experiments on RefCOCO/+/g benchmarks show improvements over coordinate-based methods under both supervised fine-tuning and reinforcement learning paradigms.

**Strengths:**

1. RefWords achieves diverse referring tasks (bbox, mask, polyline) within a pure sequence-to-sequence framework, maintaining architectural simplicity while supporting multiple output formats through a single set of learnable tokens.
2. The Offset Tokens with explicit \<DELETE\> capability provide a natural self-correction loop that mimics human spatial reasoning (propose → critique → refine). This addresses the irreversibility problem in one-shot coordinate prediction, where initial errors cannot be corrected.

**Weaknesses:**

1. The paper claims coordinate strings disrupt spatial topology, but provides no visualization or analysis to support this. There is no embedding space analysis showing that learned grid tokens preserve spatial relationships, no qualitative comparison showing where coordinate methods fail.
2. The performance gains from SFT are modest in Table 1, 2, 3. More critically, the offset tokens, presented as a key novelty for "self-correcting" spatial predictions, contribute minimally. The RL results (RefWords-R1) show larger gains but this improvement is confounded.
3. The paper provides no pure RL experiment without SFT initialization, nor any comparison of coordinate-based methods trained with RL. Without these baselines, it is impossible to determine whether the gains stem from the grid tokenization scheme itself or simply from applying RL to spatial prediction tasks in general.
4. The citation format throughout the paper appears inconsistent or incorrect—proper LaTeX citation commands (e.g., ~\citep{}) do not seem to be used.

**Questions:**

1. Why sinusoidal encoding for grid embeddings rather than learned positional encoding?
2. Why element-wise addition for fusing embeddings instead of concatenation or cross-attention?
3. The grid size (n=32) and offset granularity (m=64) appear arbitrary with no hyperparameter ablation provided.

---

### Note · Authors · 2025-11-12

I have read and agree with the venue's withdrawal policy on behalf of myself and my co-authors.